# Specificity of spiders among fear- and disgust-eliciting arthropods: Spiders are special, but phobics not so much

**Eva Landová**[1,2]☯*, **Markéta Janovcová**[1,2]☯, **Iveta Štolhoferová**[1,2], **Silvie Rádlová**[1], **Petra Frýdlová**[2], **Kristýna Sedláčková**[1], **Daniel Frynta**[1,2]

**1** National Institute of Mental Health, Klecany, the Czech Republic, **2** Department of Zoology, Faculty of Science, Charles University, Prague, the Czech Republic

☯ These authors contributed equally to this work.

* evalandova@seznam.cz

**Data Availability Statement:** The data associated with this research are available at Mendeley Data Repository (https://doi.org/10.17632/68mkyrb4n3.1).

## Abstract

To investigate a specificity of spiders as a prototypical fear- and disgust-eliciting stimuli, we conducted an online experiment. The respondents rated images of 25 spiders, 12 non-spider chelicerates, and 10 other arthropods on a fear and disgust 7-point scale. The evaluation of 968 Central European respondents confirmed the specificity of spiders among fear- and disgust-eliciting arthropods and supported the notion of spiders as a cognitive category. We delineated this category as covering extant spider species as well as some other chelicerates bearing a physical resemblance to spiders, mainly whip spiders and camel spiders. We suggested calling this category the spider-like cognitive category. We discussed evolutionary roots of the spider-like category and concluded that its roots should be sought in fear, with disgust being secondary of the two emotions. We suggested other chelicerates, e.g., scorpions, might have been important in formation and fixation of the spider-like category. Further, we investigated an effect of respondent's sensitivity to a specific fear of spiders on evaluation of the stimuli. We found that suspected phobic respondents were in their rating nearly identical to those with only high fear of spiders and similar to those with only moderate fear of spiders. We concluded that results based on healthy respondents with elevated fear should also be considered relevant for arachnophobia research.

## Introduction

Evolutionary perspective offers an explanation why ancient biological stimuli that were threatening to our ancestors have been prioritised by our category-specific visual attention (animals [1], snakes [2], spiders [3], big cats [4], human faces [5]) and why these reactions are accompanied by strong emotions to this day [6]. The neuroscientists explore complex ways in which neural circuits are involved in connecting various areas responsible for attention, perceiving fear, and motor reaction [7, 8]. These circuits enable quick reaction to a specific life-threatening stimulus and is commonly known as the fear module [9, 10].

There is no doubt that throughout the evolutionary history, many animal species have been an important source of imminent threat to our survival either as predators [11], or parasites

**Funding:** This project has been supported by the Czech Scientific Foundation (GAČR), project No. 19-07164S, awarded to EL. https://www.gacr.cz/en/ The funders had no role in study design, data collection and analysis, decision to publish, or preparation of the manuscript.

**Competing interests:** The authors have declared that no competing interests exist.

[12]. To this day, certain animals including spiders evoke high levels of fear and disgust (reviewed in [13]). In a survey using the standard Spider Phobia Questionnaire (SPQ), 10.3% out of 3 863 Czech respondents reported very high fear of spiders (scoring 22 or higher on 31-point scale; [13, 14]). Arachnophobia, an irrational, uncontrollable fear of spiders, is one of the most common specific animal phobias affecting 2.7–6.1% of general population, women significantly more often than men [15, 16]. These negative emotions associated with spiders are even more intriguing since only 0.5% of all spider species represent a real potential threat to humans [17].

Due to higher fear or even phobia of spiders being so prevalent in a general population, one could hypothesize its evolutionary roots. Spiders might have represented a real threat to our ancestors; thus, a rapid fear response would be highly adaptive. Subsequently, this specific fear of spiders (or similar invertebrates) or at least a predisposition for fast associative learning of fear response [18] would become genetically fixed through natural selection. This view is consistent with the idea of Seligman's biological preparedness [19]. Should this be the case, we can hypothesize that people share this negative attitude across cultures, although Davey [20] attributed this phenomenon to shared cultural stereotype. Moreover, spiders evoke not only fear, but high level of disgust too [21]. Specifically, Lorenz et al. [22] found that aversion toward spiders is associated with pathogen disgust. Disgust originally evolved because it served as an effective mechanism for orally rejecting harmful substances without tasting them [23]. It allowed humans to avoid the ingestion of pathogens, too [24]. Related idea posits that, in human ancestors, disgust has increased avoidance of pathogens, parasites and possible sources of contamination [25]. Different possible ways of getting infection are important for this hypothesis: infection through skin or genitals contact with surfaces, ingestion of pathogens and parasites through contamination, and contact with diseases transmitting animals [20, 26]. These two evolutionary explanations of how spiders could have become emotionally salient stimuli are not mutually exclusive.

Several lines of evidence further point toward the evolutionary roots of negative emotions elicited by spider stimuli. Among those, the most serious one seems to come from developmental studies which support the view of the spiders as an important cognitive category already in infants [27–29], some as young as 5 month old [30]. However, indirect indications can be further named. One, as mentioned earlier, in self-reports, respondents typically state that spiders evoke equally fear as well as disgust [13, 31, 32]. This testifies to the widespread negative attitude toward spiders across respondents with different educational and socioeconomical background. While the negative attitude can be contributed to a learned culture stereotype, the only cross-cultural study we know of [33] reports on comparable attitudes in South African respondents. Two, in accordance with the preparedness hypothesis [19], respondents associate fear more readily with the spider stimuli than the neutral stimuli [10, 34] and such fear is less prone to extinction [35, 36]. Moreover, similar results are reported under the instructed extinction paradigm, which involves informing participants after the fear learning, that unconditional stimulus (electric shock) will no longer be present. This method facilitates extinction in fear irrelevant stimuli, however if the fear relevant stimuli were images of snakes and spiders, the fear was not sensitive to instructed extinction [37, reviewed in 38]. Nonetheless it was also shown that acquired fear inhibition can be modulated by participants' sensitivity to fear of spiders [34] and lately, this line of argumentation has been questioned [39–41]. Three, respondents are attracted or distracted by spiders in visual attention tasks [42–44] suggesting spiders may be evolutionarily persistent threat specified for visual detection and attention capture. However, other papers show that the personal relevance of the spider stimuli is crucial [45] as well as its potential goal-relevance to the task [46]. While none of the indicators can be considered a conclusive evidence, cumulatively, they provide a reasonable

argument for investigation of potential evolutionary roots of negative emotions associated with spiders.

Emotions can direct automatic attention to emotionally salient stimuli [47], such as spiders or snakes, and sometimes even precede conscious perception [48]. However, perception as a cognitive process of transformation of proximal stimulus into a percept (the accessible, subjective experience that is connected with activation of a certain category in the mind; [49]) modulates further late attention towards evolutionary relevant threatening stimuli [50]. Michalovski et al. [51] studied temporal dynamics of visual attention to the spiders using ERPs (event related potentials). They found that the spiders are processed preferentially in later stages of perceptual and evaluative processing, especially in spider phobics. Late cognitive stimulus evaluation, like its proper categorization, is thus important when we are confronted with classes of stimuli that have (or had in our evolutionary past) direct relevance for our well-being and survival than others. We can expect that extremely relevant stimuli are categorized into special emotional categories in human mind, which may differ from other categories [49], and they are preferentially processed in the brain [52, 53]. Forming the stimulus category in the mind is thus a cognitive process when people group certain objects or concepts as equivalent or analogous reducing the information complexity, but they acquire set of information thanks to association of the object with a certain category [54]. Proper categorization of the potentially life-threatening stimuli may still direct our late attention on the one hand, but may allow for effective regulation of the impact of negative emotions like fear or disgust on the other one [55]. Categorization of emotional stimuli as a cognitive process assumes the existence of categories based on the everyday experience or evolutionary past in some cases. If people with very different experiences form similar emotional categories containing life-threatening animal stimuli like scorpions as well as harmless spiders, it may indicate the existence of a pre-existing general category for these incentives in human mind, which may be generalized to a wider group of animal species. This argument supports the hypothetical existence of evolutionarily rooted negative emotions of specific animal stimuli, similarly to a more frequently used argument of the cross-cultural agreement in emotional evaluation of these stimuli [33].

Based on this, we hypothesise that some animal stimuli may form a specific category inside the human mind on the basis of shared morphological features perceived via our sensory system. Such cognitive category can additionally interact with emotional processing during its perception. Therefore, forming a cognitive category goes along with emotional evaluation, making it the cognitive process.

Are spiders therefore perceived as a specific group distinct from other invertebrates? Gerdes et al. [21] compared subjective emotional evaluation of spiders and three other groups of insects: beetles, bees and wasps, butterflies, and moths. They found that spiders evoke more fear and disgust than the other groups and they concluded that among these groups, spiders are truly specific stimuli. Contrary, Breuer et al. [56] found that all crawling invertebrates, spiders included, are perceived more negatively compared to those that can fly by 9–13 years old children. Shipley and Bixler [57] offered US college students 10 silhouettes of insects, spiders, and other invertebrates in paired forced choice test. In this study, spiders formed one cluster together with a praying mantis, wheel bug, stag beetle and a house centipede. Despite great attention paid to the study of fear and disgust evoked by spiders [13, 20, 32, 51, 58–61], the question of specificity of spider stimulus still remains open.

For these reasons, general aim of this study is to determine prototypical stimuli (spiders and spider-like arthropods) that elicit pronounced emotional response. We asked whether high negative emotional evaluation (fear and/or disgust) is specific to spiders compared to other arthropods. Regarding phylogeny, spiders are representatives of Chelicerata which in turn are one of four major extant arthropod groups (other three being Myriapoda, Crustacea

and Insecta; for detailed phylogeny and taxonomy see [62], and S1 Table in S1 File). To answer our questions, we chose a wide variety of stimuli including several representatives of spiders, nine other main clades of chelicerates as well as representatives of above-mentioned arthropod groups. Together, the selected stimuli represent full morphological diversity of living spiders and its closest relatives allowing for a precise comparison on a very fine scale. Further, we asked which morphological features of spiders are responsible for their emotional evaluation. Because the emotional evaluation of spiders is closely related to the respondents' sensitivity to a specific fear of spiders [13, 21], we tested people with normative as well as high fear of spiders. We focused on covering a full spectrum of respondents from those with low or no fear of spiders to suspected phobic and near phobic respondents. Relatively large numbers of diverse respondents are firstly crucial for validly defining spiders as a prototypical stimulus in a general population. Secondly, it allows investigating from what point specific fear of spiders affects subjective emotional evaluation of spider and spider-like stimuli in a manner a simple comparison of two extreme groups from opposite sides of the "fear spectrum" cannot.

The particular questions and aims of this paper are as followed. (1) Is position of taxonomically defined spiders on fear and disgust scales distinctive compared to that of other chelicerates and arthropods? (2) Do spiders form a single distinct cognitive category or more species of invertebtrates are perceived as a "spider"? (3) Which spider morphotypes are associated with fear and/or disgust rating of the stimulus? (4) Which characteristics of the respondents are predictors of fear and disgust rating of spiders and other arthropods? (5) Is there a systematic difference in ratings of suspected phobic respondents compared to those with low, moderate, and high fear of spiders?

## Methods

### Participants

All respondents were adult Czech, Slovak, or other Central Europeans, aged 18 to 77 (median = 29, mean = 30.86, SD = 11.01), both men and women. All respondents were tested completely online via a special web application [63, 64]. Respondents were actively recruited via promoting the research on Facebook sites of involved institutions (i.e., Faculty of Science, Charles University; National Institute of Mental Health; Faculty of Science, University of South Bohemia in České Budějovice), of Facebook influencers, and by advertising to respondents of Human Ethology Research Group. The recruitment and testing took place during spring 2020.

Out of 968 respondents who rated all the stimuli, 704 (72.73%) were women and 264 (27.27%) were men. The women to men ratio was stable during all phases of testing. Out of these 968 respondents, the vast majority of 875 (90.39%) respondents filled in three additional questionnaires, which were used to further characterize the respondents. These were (1) the Spider Phobia Questionnaire (SPQ; [65]), (2) the Mini-spider (a questionnaire asking about respondent's attitude toward spiders and traits associated with their potential fear and/or disgust of spiders, see S2 Table in S1 File for full version of the questionnaire) and (3) the Disgust Scale-Revised (DS-R; developed by Haidt et al. [66]; modified by Olatunji et al. [67]; translated to Czech by Polák et al. [68]). For purpose of statistical analyses, we divided respondents based on their SPQ score. The limits of defined categories were chosen as they correspond to 25th, 50th, 75th, and 90th percentile of SPQ scores assessed from an independent sample of Czech respondents (N = 3863; [13, 14]). Categories were as follow: SPQ score 0–2 –extremely low fear respondents, 3–6 –low fear respondents, 7–15 –moderate fear respondents, 16–22 –high fear respondents, 23–31 –suspected phobic respondents. Out of 875 respondents were 186 of the extremely low fear (21.26%), 216 of low fear (24.69%), 214 of moderate fear (24.46%), 170

of high fear (19.43%), and 89 of suspected phobic (10.17%) category. This method was previously used for Snake Questionnaire (SNAQ; see [69]). In our sample, there were slightly more respondents with SPQ > 15 than expected (25% expected, 29.60% observed). We hypothesize that this is a result of recruiting the respondents online, as the fear of spiders related research is more attractive to people who indeed fear spiders. A vast majority of these respondents were women– 244/259, i.e., 94.21%. This is in accordance with the fact that women show generally higher tendency to experience negative emotions such as fear [70, 71]. Furthermore, LeBeau et al. [72] showed that up to 91% of animal phobics were women.

## Selection and preparation of stimuli

We prepared a set of 47 pictures each representing one species of Arthropods. Since our focus was on spiders, about a half of the pictures (25/47) were spider species (Aranea). The species were selected to represent the fundamental diversity of spider morphology (for a phylogenetic tree of spiders, see [73]) although no species with adults under 5 mm of body length were used. About a quarter of pictures (12/47) depicted other chelicerate species, close relatives to spiders. These were once again chosen to represent a full diversity of chelicerate morphology. The rest of the pictures (10/47) depicted other arthropods, such as crustaceans, insects, millipedes, and centipedes. Species of this group were chosen based on their morphological resemblance to spiders, e.g., long thin legs (a water measurer *Hydrometra stagnorum*, a common hermit crab *Pagurus bernhardus*), an overall body shape (a crab *Liocarcinus vernalis*, a swift lousefly *Crataerina pallida*), multiple legs (a common striped woodlouse *Philoscia muscorum*, the millipede species) or potential dangerousness (a venomous centipede *Ethmostigmus trigonopodus*, a common earwig *Forficula auricularia* rumoured to crawl into and infest people's ears). For a full list of used stimuli, see Table 1 and S1 Table in S1 File. For each selected species, we found a representative photograph of an adult individual on the Internet (Flickr or Wikimedia Commons, both licensed under the Creative Commons license). Only photos in suitable resolution (at least 800 x 533 pixels) depicting the animal in full body were chosen. We adjusted the photos to a standardized form by placing the animals on a white background and into a similar position and comparable body size, see Fig 1.

## Testing procedure

A total of 968 respondents evaluated the set separately for perceived fear and disgust on a seven-point Likert scale (1 standing for the lowest fear or disgust, 7 standing the strongest fear or disgust; [74]). Before the evaluation, each respondent filled in a short questionnaire concerning the age, gender, level of education (2 levels—university and lower), type of education (4 levels–biology, medicine, engineering and other technical fields, and other), personal attitude toward spiders (scale 1–7, 1 for very positive, 7 for very negative), and frequency of encountering spiders (scale 1–3, 1 for rarely, 3 for often). Further, he or she was informed about the content of the experiment and provided his/her consent to the processing of personal data, all in the Czech language. A total number of 1,513 respondents started the testing procedure, however only 968 (63.98%) respondents rated all of the stimuli by both emotions. The order in which were pictures presented was randomized separately for each respondent. The stimuli order for fear evaluation and for disgust evaluation were generated independently, 478 of respondents rated the stimuli by perceived fear first and 490 by perceived disgust first. Out of these 968 respondents, the vast majority of 875 (90.39%) respondents filled in three additional questionnaires (SPQ, the Mini-spider, and DS-R). The experiment as presented to respondents is available through the following link (the English language version): https://www.krasazvirat.cz//sets/?set=53&lang=1.

**Table 1. Mean scores and their standard deviations (SD) for all stimuli as rated by all respondents (N = 968).**

| Stimulus | Group (Clade) | Mean (SD) for Fear | Mean (SD) for Disgust |
|---|---|---|---|
| *Aphonopelma eutylenum* | Spider | 4.774 (2.072) | 4.379 (2.193) |
| *Aptostichus miwok* | Spider | **5.204** (1.839) | **5.046** (1.928) |
| *Araneus diadematus* | Spider | 4.696 (1.959) | 4.647 (2.051) |
| *Cheiracanthium inclusum* | Spider | 4.681 (1.907) | 4.718 (1.971) |
| *Eratigena atrica* | Spider | 4.729 (2.043) | 4.621 (2.115) |
| *Falconina gracilis* | Spider | 4.561 (1.928) | 4.441 (1.997) |
| *Grammostola porteri* | Spider | 4.769 (2.104) | 4.392 (2.311) |
| *Latrodectus mactans* | Spider | **5.366** (1.818) | 4.794 (2.033) |
| *Lyssomanes viridis* | Spider | 4.312 (1.985) | 4.093 (2.083) |
| *Macrothele taiwanensis* | Spider | **5.101** (1.884) | **4.925** (1.987) |
| *Maevia inclemens* | Spider | 4.604 (2.173) | 4.454 (2.204) |
| *Maratus speciosus* | Spider | 4.084 (2.297) | 3.555 (2.261) |
| *Mecaphesa dubia* | Spider | 4.751 (1.955) | 4.619 (2.027) |
| *Miagrammopes flavus* | Spider | 4.258 (1.947) | 4.066 (1.979) |
| *Micrathena schreibersi* | Spider | 5.030 (1.899) | 4.664 (2.042) |
| *Myrmaplata plataleoides* | Spider | 4.015 (1.835) | 4.270 (1.910) |
| *Nephila pilipes* | Spider | 4.501 (1.975) | 4.387 (2.055) |
| *Oxyopes macilentus* | Spider | 4.603 (1.937) | 4.493 (2.044) |
| *Phidippus texanus* | Spider | 4.310 (2.229) | 4.043 (2.297) |
| *Pholcus phalangioides* | Spider | 3.296 (2.126) | 3.561 (2.157) |
| *Rabidosa rabida* | Spider | 4.642 (2.047) | 4.491 (2.115) |
| *Salticus scenicus* | Spider | 4.349 (1.906) | 4.238 (1.988) |
| *Steatoda nobilis* | Spider | 4.728 (1.973) | 4.569 (2.021) |
| *Tasmanicosa leuckarti* | Spider | 5.075 (1.926) | 4.836 (2.065) |
| *Theraphosa blondi* | Spider | 5.083 (1.995) | 4.654 (2.151) |
| *Ammothea hilgendorfi* | Other Chelicerate | 4.582 (2.001) | 4.851 (1.969) |
| *Centruroides vittatus* | Other Chelicerate | 4.527 (1.927) | 3.636 (1.926) |
| *Cryptocellus goodnighti* | Other Chelicerate | 4.785 (1.897) | 4.727 (1.943) |
| *Gluvia dorsalis* | Other Chelicerate | 4.796 (1.815) | 4.856 (1.848) |
| *Hubbardia briggsi* | Other Chelicerate | 4.229 (1.768) | 4.518 (1.756) |
| *Hypoctonus gastrostictus* | Other Chelicerate | 4.638 (1.841) | 4.675 (1.875) |
| *Ixodes pacificus* | Other Chelicerate | 4.101 (2.007) | 4.622 (1.977) |
| *Ortholasma levipes* | Other Chelicerate | 4.506 (1.958) | 4.558 (1.934) |
| *Phalangium opilio* | Other Chelicerate | 3.786 (2.158) | 4.036 (2.138) |
| *Phrynus parvulus* | Other Chelicerate | 4.925 (1.957) | 4.818 (2.011) |
| *Roncus lubricus* | Other Chelicerate | 4.502 (1.939) | 4.707 (1.905) |
| *Trombidium holosericeum* | Other Chelicerate | 3.971 (1.921) | 4.735 (1.918) |
| *Liocarcinus vernalis* | Other Arthropod | **2.542** (1.786) | **2.557** (1.802) |
| *Pagurus bernhardus* | Other Arthropod | **2.100** (1.616) | **2.253** (1.714) |
| *Philoscia muscorum* | Other Arthropod | 2.723 (1.823) | 3.460 (1.977) |
| *Crataerina pallida* | Other Arthropod | 3.723 (1.873) | 4.268 (1.874) |
| *Forficula auricularia* | Other Arthropod | 2.954 (1.882) | 3.601 (1.994) |
| *Hydrometra stagnorum* | Other Arthropod | **2.507** (1.737) | **2.749** (1.857) |
| *Cryptops anomalans* | Other Arthropod | 3.604 (2.015) | 4.351 (2.015) |
| *Ethmostigmus trigonopodus* | Other Arthropod | 4.452 (1.953) | **5.005** (1.929) |
| *Scutigera coleoptrata* | Other Arthropod | 3.705 (2.002) | 4.405 (1.997) |

*(Continued)*

**Table 1.** (Continued)

| Stimulus | Group (Clade) | Mean (SD) for Fear | Mean (SD) for Disgust |
|---|---|---|---|
| *Spirobolida* | Other Arthropod | 3.167 (2.081) | 3.879 (2.201) |

In fear, the spider species scored the highest–*Latrodectus mactans* (5.366), *Aptostichus miwok* (5.204) and *Macrothele taiwanensis* (5.101). Similarly, in disgust, *A. miwok* (5.046) and *M. taiwanensis* (4.925) again received the first and third highest scores, respectively. However, myriapod *Ethmostigmus trigonopodus* scored the second highest in disgust (5.005). In both emotions, the crabs *Liocarcinus vernalis* (2.542 fear; 2.557 disgust) and *Pagurus bernhardus* (2.100 fear; 2.253 disgust), and the aquatic bug *Hydrometra stagnorum* (2.507 fear; 2.749 disgust) scored the lowest. Species with the three highest and lowest rankings in either emotion are in bold.

## Extraction of stimuli characteristics

For further analyses, we characterised the colouration and certain morphological traits of the stimuli. In terms of the colouration, we extracted the pixel values of each photo in the hue-saturation-lightness (HSL) colour space using the software Barvocuc [75] and following the method described in [76]. The colour value was extracted for black, white, grey, yellow, red, blue, green, brown/orange, pink, and purple and represents the number of pixels of each colour in the photograph. Additionally, we extracted average lightness and saturation of each stimulus. Regarding morphology, the investigated traits were body length, body width, leg length, leg width, body area, body perimeter and eye diameter as measured on the photographs. Since the size of the photographs was the same, the measurements were taken in pixels. Lastly, three independent observers sorted all the stimuli in five groups based on the amount of hair covering the body of the stimuli (1 for no hair, 5 for the most hair); this evaluation was used to express the level of hairiness.

## Data analyses

In order to quantify an amount of agreement in species rating provided by different respondents, we computed the two-way random, single score consistency intraclass correlations (standard ICCs). To quantify the congruence on mean values, we adopted the two-way random, average score consistency intraclass correlations (ICCs for averages; [77, 78]). A Pearson correlation coefficient was further calculated between the mean fear and disgust ratings. To further characterize data, we computed means of fear and disgust rating of each stimulus. We tested means of different stimuli using post hoc Friedman-Neményi test. Next, we computed means of fear and disgust rating of each stimulus by respondents of each SPQ category (see Discussion section). We used Spearman rank correlation of stimuli mean scores to compare ratings of different categories of respondents.

To visualize the structure of the data sets, we used cluster analysis based on raw data. We performed cluster analysis separately for each investigated emotion, distance matrix was calculated as 1-Pearson correlation among species ratings, tree diagrams were calculated using Ward's method. We used factor analysis (FA; principal component extraction method) based on correlation matrix to assess multivariate relationships among stimuli and to extract uncorrelated axes for further analysis. To determine the number of factors retained for the analysis, we used parallel analysis [79]. We performed FA for both investigated emotions together and for each dataset separately, Varimax normalized rotation was used in both cases. When analysed together, same stimuli rated by both fear and disgust always contributed to the same factor. The only exception was two separate factors, one best correlated with (only) fear ratings and the other with (only) disgust ratings of the same stimuli. Therefore, factors were mainly determined by stimuli identity and no "interaction" between the stimulus identity and its rated emotion was found. When each dataset was analysed separately, factors showed a similar

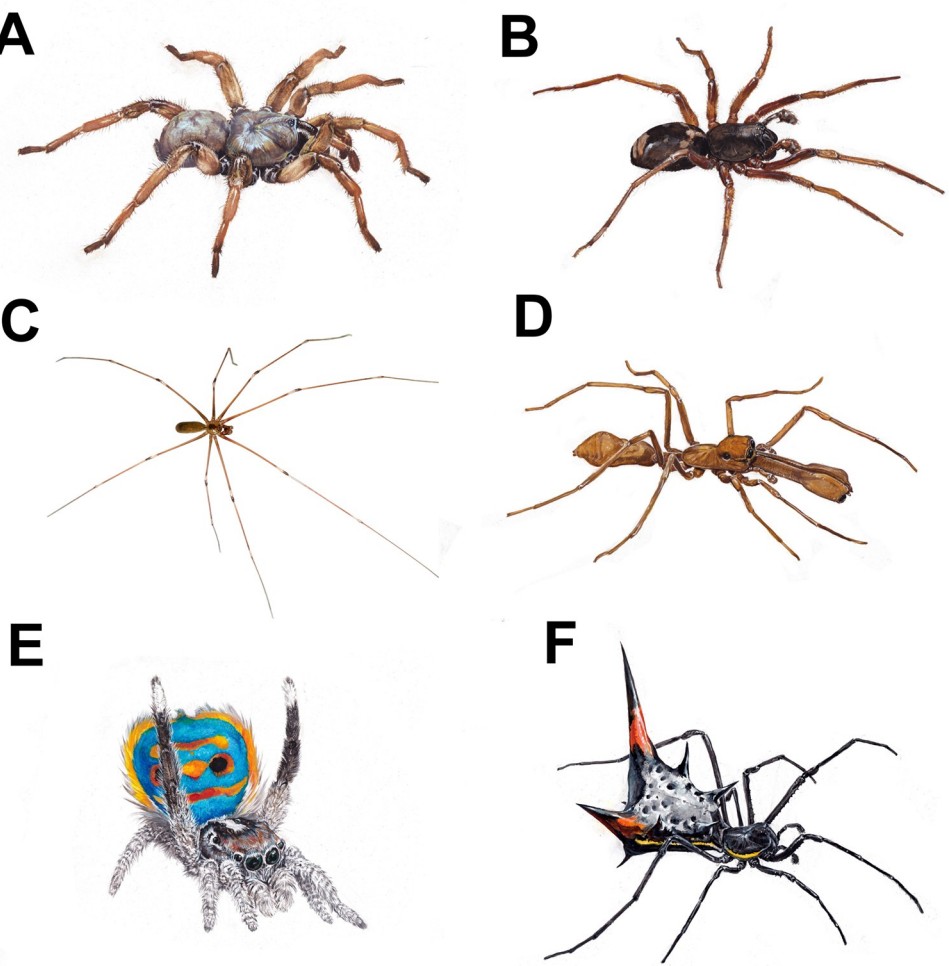

**Fig 1.** Illustrational pictures showing the variability of the presented spider stimuli (however very similar but real photos depicting the species were presented as the real stimuli): *Aptostichus miwok* (A), *Falconina gracilis* (B), *Pholcus phalangioides* (C), *Myrmaplata plataleoides* (D), *Maratus speciosus* (E), and *Micrathena schreibersi* (F). Authors: A-B, D-F–MVDr. Pavel Procházka; C–David Short.

structure as in the joined analysis. We found the approach of each emotion analysed separately fitting better with the aim of this study, hence, we decided to continue with results of the separate analyses. Computed factor scores were used as response in ANOVA testing of SPQ categories (see Discussion section). We employed post hoc Tuckey HSD for unequal N test to correct for multiple testing.

For extracting constrained gradients of variability in fear and disgust ratings, we used a redundancy analysis (RDA) [80]. Firstly, RDA was used to access contributions of certain morphological traits of the stimuli associated with their fear or disgust ratings. The entered traits were as listed in previous section (Extraction of stimuli characteristics), only stimuli from spider group and other chelicerates group were investigated. This was because rather extreme morphotypes of some stimuli from other arthropods group (mainly the centipedes) obscured the analysis and overshadowed gradients on finer scale. Secondly, we performed RDA constrained by characteristics of respondents. In full models, investigated variables were respondent's gender, age, type of education, level of education, personal attitude toward spiders,

frequency of encountering spiders, first rated emotion, and the scores of SPQ and DS-R questionnaires.

Software R [81] was used for computation of ICC (irr package) [82], Friedman-Neményi test (PMCMR package) [83], parallel analysis (EFA.dimensions package) [79], and RDA analysis (vegan package) [84]. Software Statistica 9.1. [85] was used to extract mean ratings of the stimuli, and to perform Pearson and Spearman correlations, factor analyses, cluster analyses, and ANOVA with post hoc Tuckey HSD for unequal N tests. Software Barvocuc [75], GIMP 2.10.15 (https://www.gimp.org/), Image Tool 3.1 [86], and Image J 1.40g [87] were used for preparation and characterization of the stimuli.

Full datasets associated with this study are available in Mendeley repository under the link: https://doi.org/10.17632/68mkyrb4n3.1.

### Ethical note

All procedures performed in this study were carried out in accordance with the ethical standards of the appropriate institutional research committee (the Ethic Commission of National Institute of Mental Health, approval no. 117/18, granted on 28 March 2018), and with the 1964 Helsinki declaration and its later amendments or comparable ethical standards. Written informed consent was obtained from all participants included in the study.

## Results

### Agreement among the respondents

First, we computed standard ICCs to quantify an amount of agreement in rating of the stimuli among the respondents. The ICC values computed for all respondents were relatively low, 0.239 (95% CI = 0.178, 0.332) and 0.167 (0.122, 0.241) for fear and disgust, respectively. For a subset of high fear and suspected phobic respondents (i.e., 259 respondents with SPQ score > 15) corresponding values were slightly higher 0.412 (95% CI = 0.325, 0.526) and 0.316 (0.241, 0.422) for fear and disgust, respectively. The relatively low amount of agreement among respondents suggests in turn relatively high variability in respondents' ratings. This remaining component of variation is large enough to be examined by multivariate methods.

Next, we computed ICCs for averages which indicated an accuracy of calculated total mean rating of each stimulus. These values were very high (0.997 and 0.995 for fear and disgust, respectively), which is critical for further comparisons among the mean ratings of all 47 stimuli.

### Mean rating of individual stimuli

Mean rating of examined stimuli according to elicited fear and disgust are given in Table 1. Spiders and other chelicerates tend to score high in fear, while this pattern is not so clear in the case of disgust. In both emotions, crustaceans and insects tend to score the lowest. Post hoc Friedman-Neményi test showed that a majority of comparisons among the stimulus means was significant (see S3 and S4 Tables in S1 File for details). This means that the respondents differentiate well even among the stimuli belonging to the same group (clade).

A comparison of mean fear and disgust scores of the same stimuli revealed high correlation between fear and disgust scores (r = 0.864, P < 0.0001). To adjust for this, we computed differences between mean fear and disgust ranking for all stimuli. The results are shown in Fig 2. Notice that a vast majority of spiders exhibit higher fear score than disgust score. Contrarily, a vast majority of the rest of the stimuli has higher disgust score when compared to their fear score.

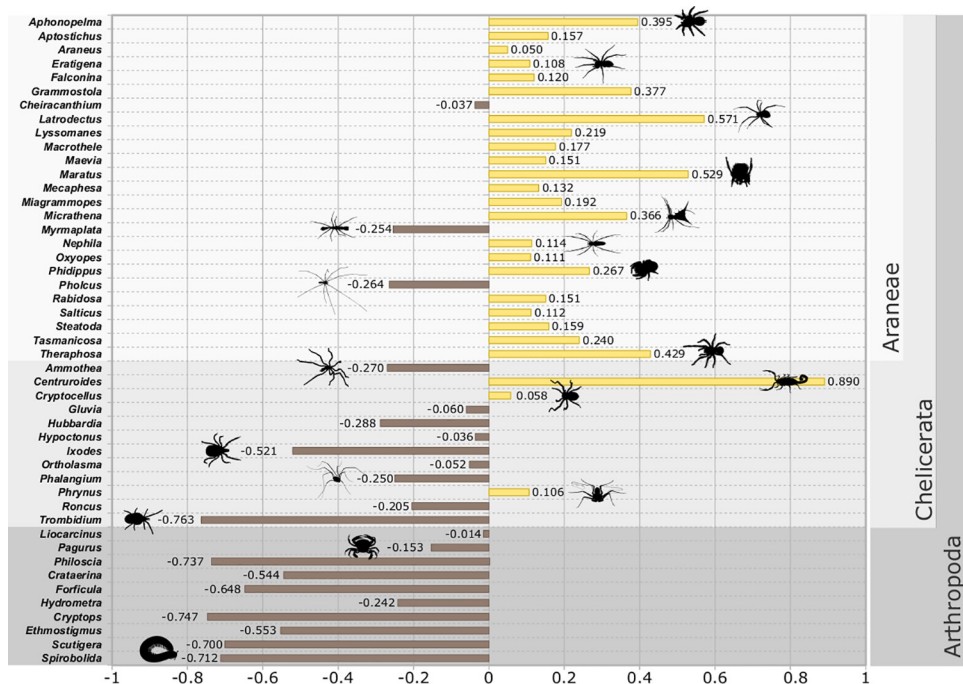

**Fig 2. The difference between the mean fear score and mean disgust score for each examined stimulus.** Notice that the vast majority of spider stimuli have higher mean fear score than disgust score (yellow bars, right side of the figure) while the vast majority of other chelicerates and other arthropods have higher mean disgust score than fear score (brown bars, left side of the figure).

## Multivariate analyses

Multivariate structures of the fear and disgust datasets are visualized by results of cluster analyses. In both cases, spiders and spider-like chelicerates (further referred to as "spider cluster") were clearly separated from the rest of the stimuli (forming "non-spider cluster"), i.e., myriapods, crustaceans, insects and the rest of chelicerates (including a scorpion and a tick). The only exceptions represent a myrmecophilous spider *Myrmaplata plataleoides* and a whip scorpion *Hypoctonus gastrostictus*. These two stimuli cluster together with spiders according to fear but fall into non-spider cluster according to disgust. Spider cluster further splits into three subclusters: (1) robust hairy spiders, (2) spider-like chelicerates and (3) gracile spiders (Fig 3).

In order to perform unconstrained gradients in fear and disgust ratings, we introduced FA. We identified three (fear dataset) and four (disgust dataset) factors using parallel analysis (see S5 Table in S1 File for results of parallel analysis). For fear, extracted factors explained 33.67, 18.95 and 16.00% of variance, respectively (68.62% in total). The corresponding values for disgust were 25.85, 18.48, 23.35 and 5.08% (72.76% of explained variance in total). For both emotions we interpret the first three factors as (1) fear/disgust of general spiders, (2) fear/disgust of non-spiders, and (3) fear/disgust of hairy spiders. The fourth factor extracted from disgust dataset corresponds to crabs (for factor loadings and other details of FA, see S6 Table in S1 File for results of FA performed separately for each emotion and S7 Table in S1 File for results of the combined FA).

## Contributions of stimuli characteristics

To investigate the effect of morphological features on stimuli ratings, we employed RDA. We included only chelicerate stimuli, as other arthropods would obscure the analysis due to their

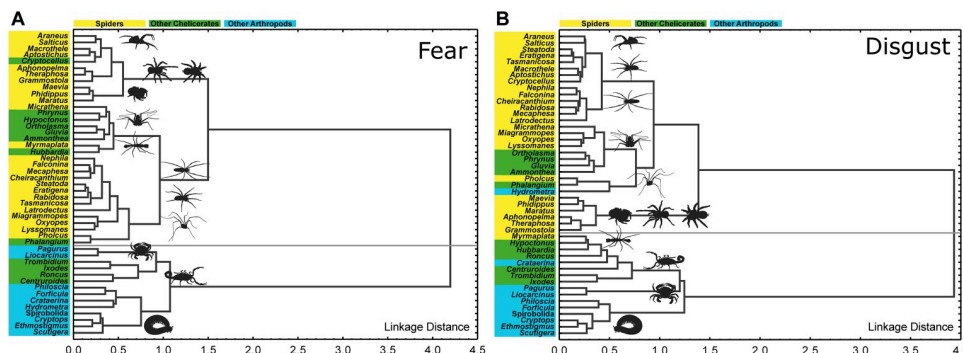

**Fig 3.** Cluster trees based on fear (A) and disgust (B) scores. In both cases, spiders and spider-like chelicerates form a clearly separate cluster from the rest of the stimuli, i.e., myriapods, crustaceans, insects, and the rest of chelicerates (including the scorpion and tick). Spider cluster further splits into three subclusters: robust hairy spiders, spider-like chelicerates, and gracile spiders.

different body plan. RDA model explained 34.1% and 37.8% for fear and disgust, respectively (S8 and S9 Tables in S1 File). For both emotions, spider stimuli were located along one main gradient interpretable as robustness versus gracility. It was formed by body area (for fear + black colour; for disgust + eye diameter) on one side and body perimeter (for disgust + leg length) on the other side. Body length and red colour (both high in non-spider chelicerates) contributed to the other axis. The results are shown in S1 Fig.

## Interindividual differences of rating

We employed RDA to extract constrained gradients from fear and disgust ratings. In analysis that was constrained for the respondents' characteristics, the final model explained 35.82 and 35.50% of variability for fear and disgust, respectively (S10 and S11 Tables in S1 File). In both datasets, SPQ, Personal attitude toward spiders, and DS-R contributed positively, while Gender, Age and Biological type of education contributed negatively to the first axis. We interpret the first axis as negative response to the evaluated stimuli, in particular spiders. Only two variables, biological education (positively) and DS-R (negatively) contributed considerably to the second axis, which separates 'spider' and 'non-spider' clusters (Fig 4). In disgust but not fear, the factor of the first-rated emotion was associated with negative rating of stimuli. While significant, the effect was very small (S11 Table in S1 File). As illustrated by RDA results, SPQ scores themselves predicted mean rating given to spider stimuli by a given respondent: $r_{spearman} = 0.778$ and $0.771$ for fear and disgust, respectively.

## Low-fear and high-fear respondents

We split respondents according to SPQ values to compare their rating of individual stimuli: (a) 0–2 –extremely low fear, (b) 3–6 –low fear, (c) 7–15 –moderate fear, (d) 16–22 –high fear, and (e) 23–31 –suspected phobic respondents (see Methods: Participants section for details). Total mean scores and agreement among the respondents increased gradually with SPQ categories for both examined emotions (S12 and S13 Tables in S1 File). Mean scores per stimulus given by a subset of suspected phobic respondents were highly correlated with scores of high fear respondents (Spearman rank correlation of stimuli mean scores), $r_{spearman} = 0.976$ and $0.959$ for fear and disgust, respectively. Further, mean scores given by both of these groups of respondents were poorly correlated (fear) or uncorrelated (disgust) with means of extremely low fear respondents (S14 Table in S1 File). We examined this pattern using ANOVA with SPQ categories as explanatory variable and factor scores derived from FA (see above) as

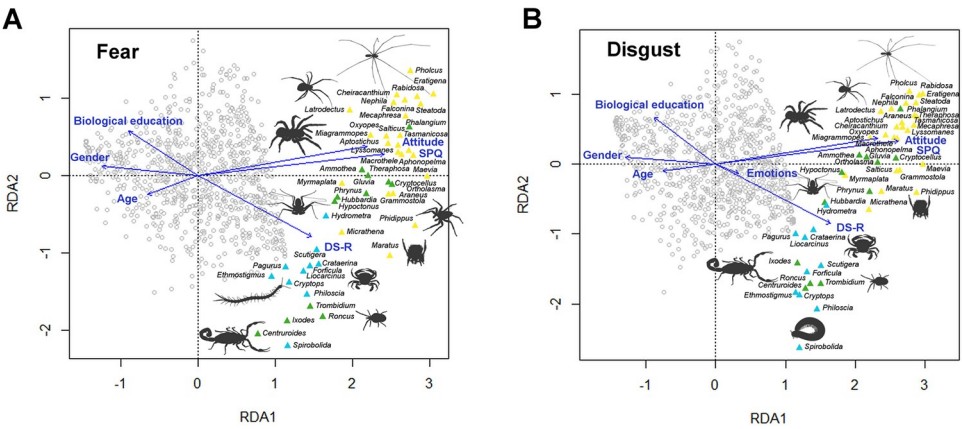

**Fig 4.** Results of the RDA analysis for fear (A) and disgust (B) scores. Spider stimuli are in yellow, other chelicerates in green, and other arthropods in blue; grey circles stand for respondents. Blue arrows represent variables entered the analysis ("Attitude" for the attitude toward spiders; "Emotions" for the first rated emotion).

response. Factors rather than phylogeny-based classification were used as they represent cognitive categories as assessed through the respondents' ratings. We employed post hoc Tuckey HSD for unequal N test to correct for multiple testing. Mean estimates of Factors 1 and 3 (representing fear/disgust of spiders and robust hairy spiders, respectively) were significantly different for all between-group comparisons but one–high fear respondents and suspected phobic respondents did not differ in their rating of spider stimuli. Contrarily, mean estimates of Factor 2 (representing fear/disgust of other arthropods) were similar for all groups of respondents with only the extremely low fear respondents differing from most of the other groups. This pattern holds true for analysis based on both fear and disgust ratings (S15 Table in S1 File) and is illustrated in Fig 5.

## Discussion

### Is position of taxonomically defined spiders on fear and disgust scales distinctive compared to that of other chelicerates and arthropods?

When examining the mean ratings of stimuli, spiders ranked among the highest of all species according to both fear and disgust (see Table 1). All but four (out of 25) species of spiders

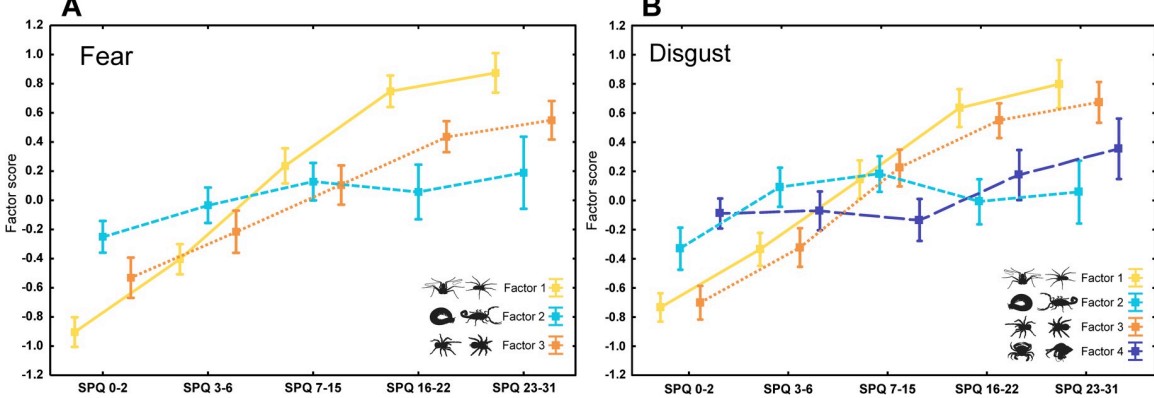

**Fig 5. Means and standard error bars of factor scores as extracted from FA for each SPQ category of respondents.** In both fear (A) and disgust (B), Factor 1 roughly corresponds to fear/disgust of gracile spiders and spider-like chelicerates, Factor 2 to fear/disgust of other arthropods, Factor 3 to fear/disgust of hairy robust spiders, and Factor 4 to disgust of crabs.

ranked above average score of the stimuli altogether. Similarly, 18 spider species scored above average in disgust. Among the top ten highest ranked species according to fear, 7 were spiders. They were the southern black widow *Latrodectus mactans* (ranking at the very top), tarantula species of the genera *Aptostichus*, *Macrothele*, *Theraphosa*, and *Aphonopelma*, the strangely looking orb-weaver spider *Micrathena schreibersi* with extremely long spines serving for anti-predator defence [88], and the wolf spider *Tasmanicosa leuckartii* characteristic by its relatively large size. Three remaining species were arachnids highly resembling spiders in appearance–the whip spider *Phrynus parvulus*, camel spider *Gluvia dorsalis*, and the hooded tickspider *Cryptocellus goodnighti*. Although these three species might look dangerous, they are harmless to humans [89–91]. On the disgust scale, the situation was similar with *Aptostichus*, *Macrothele*, *Gluvia*, *Tasmanicosa*, *Phrynus*, *Latrodectus*, and *Cryptocellus* which all scored among the top ten. Second top ranked the centipede *Ethmostigmus trigonopodus*, sea-spider *Ammothea hilgendorfi* and the mite *Trombidium holosericeum* ranked at fifth and ninth place, respectively. A parasitic tick *Ixodes pacificus* scored quite low in disgust which was surprising as animals associated with dirt, decay, or disease (e.g., worms, lice, tapeworms, or cockroaches) usually trigger high disgust [13, 92]. In our previous study, an engorged tick and other parasites elicited stronger disgust than spider picture stimuli [93]. We hypothesize that either respondents did not recognize the stimulus (we used a starved tick in the current study), or multiple spider stimuli overshadowed the disgust elicited by a single tick.

When comparing mean fear and disgust ratings of the same stimulus, a clear pattern emerged. A vast majority of spiders scored higher in fear than in disgust, while the reverse was true for a vast majority of other chelicerates and arthropods (see Fig 2). An important exception was the striped bark scorpion (*Centruroides vittatus*), which scored very low in disgust but high in fear and hence had the highest difference between its fear and disgust ranking of all the examined stimuli. Parasites (the tick and the mite) scored much higher in disgust than fear, alongside with all centipedes, the millipede of the order Spirabolida, and the woodlouse *Philoscia muscorum*. To summarize, spiders elicit both strong fear and strong disgust. Further, fear elicited by spider stimuli is stronger than disgust elicited by the same stimulus. The reverse is true for other chelicerates and arthropods. While there are exceptions to these rules, it can be concluded that based on fear and disgust ratings, spiders (Araneae) are distinct stimuli among other examined invertebrates.

## Do spiders form a single distinct cognitive category or more species of invertebtrates are perceived as a "spider "?

Although spiders are distinctive in their fear and disgust rating among other invertebrates, it does not automatically mean that they form a single distinctive cognitive category. The grounds for categories may be determined by factors related to the perceiver (e.g., fear and disgust sensitivity of the respondents, negative experience with spiders, shared evolutionary past) as well as those features inherent to the stimulus (e.g., body plan with multiple legs, chelicerae, dangerously looking appendages, hairs, thorny protrusions). Theoretically, all invertebrate species that evoke fear or disgust of a certain level may be categorized together on the basis of emotional percept only, even though they are perceptually diverse (for a review, see [94]). However, this was not the case in our study.

Both cluster analysis and factor analysis divided stimuli into two major and well-defined groups that can be characterized as a "spider cluster" and "non-spider cluster". Consistently, no matter the analysis (cluster or factor analysis) or evaluated emotion (fear or disgust), clusters were as follows. The spider cluster was formed by all but two spider species, together with the whip spider, camel spider, sea spider, hooded tickspider, and both

harvestman species (*Ortholasma levipes* and *Phalangium opilio*). The non-spider cluster was formed by an earwig *Forficula auricularia*, a lousefly *Crataerina pallida*, the millipede, all centipedes, and all crustaceans together with the scorpion, a pseudoscorpion *Roncus lubricus*, the tick, and the mite. Although the position of a few species changed among clusters depending on the analysis or dataset, the overall pattern was very stable all-across (see Fig 3). To summarize, all chelicerates similar to spiders joined one category with them, while dissimilar morphotypes were excluded. This result is consistent with the view of inherited cognitive category of emotionally salient stimuli–"spiders"–which humans have shared on the basis of coevolution [44]. However, this category can be established on the basis of perceptual similarity as well [95].

To elucidate possible evolutionary roots of spider cognitive category, we confronted our results with developmental studies. Preschool children have enhanced visual detection of spiders over the mushrooms and cockroaches [27]. Further, 6-month-old children react to spiders (and snakes) by increased pupillary dilatation which indicates increased emotional reaction compared to their reaction to flowers and fishes [28]. Even 5-month-old infants have basic perceptual template for spiders as Rakison and Derringer [30] showed in a series of experiments with simplified schematic pictures of spiders. Scrambled schematic pictures did not work compared to ones with spider features in a biologically relevant position. These schematic simplified pictures of spiders were also generalized to real photographs of spiders in habituation experiment. As the infants did not have much experience with real spiders at this age, we can assume that 5-month-old infants have innate perceptual template for threatening biological spider-like stimuli. All these results support "spiders" as an inherited cognitive category shared by humans on the basis of coevolution.

Nevertheless, it should be stressed that "spiders" as a cognitive category are not identical with spiders in a biological (taxonomical) sense, i.e., with the order Araneae. The "spiders" as a category arising from the subjective emotional evaluation of diverse arthropod species is formed by stimuli's morphological similarity to a typical spider morphotype that causes perceptual similarity for respondents. Morphologically similar chelicerates are considered spiders (e.g., the whip spider, the camel spider). Contrarily, some spiders far from a prototypical spider morphotype (e.g., the myrmecophilous genus *Myrmaplata*) can exceptionally be considered as non-spiders. In this sense, a "spider-like" cognitive category might be more convenient label. Lastly, not all spiders are alike. Two separate "spider-like" subcategories can be identified–(1) gracile, small-bodied, long-legged, smooth spiders and other chelicerates and (2) robust, large-bodied, hairy spiders roughly corresponding to tarantulas (Fig 3).

## Which spider morphotypes are associated with high fear and/or disgust rating of the stimulus?

Analysis of chelicerate morphotypes provided same results when based on fear as well as disgust. Spider species were clearly placed alongside a gradient defined by body perimeter on one side and body area on the other. Therefore, one end represented gracile species with large perimeter but small area (e.g., a long-bodied cellar spider), and the other end robust species with large area and relatively small perimeter (e.g., various tarantula species). This further supported results discussed in previous section. Robust species proved as highly salient stimuli and were those scoring high in both fear and disgust. Larger body length and higher proportion of red colour were also associated with high fear and disgust score but were driven primarily by other chelicerate species, mainly the scorpion (of very elongated body) and the mite (of dark red colour).

There is only one group of truly dangerous spider species that could have been important in the evolutionary context–the black widows (genus *Latrodectus*, family Theridiidae). Black widows are distributed in multiple continents including Africa and the Middle East [17], the area critical for coevolution with humans, and therefore they could have been an important life-threatening stimulus to our ancestors. However, this spider genus is not robust at all. There are some robust venomous spiders that might be dangerous to humans. For example, the Australian funnel-web spiders (Atractidae) have a specific neurotoxin to deter marsupial, bird, and lizard predators, however its toxicity for humans is only a coincidence from the evolutionary point of view [96]. The same is true for tarantulas (Theraphosidae) as species dangerous to humans inhabit Southern America and Australia [97] and therefore are not relevant in the evolutionary context. Accordingly, it was the black widow which scored as the most fear-eliciting stimulus. For these reasons, tarantula species should not be viewed as a core prototypical spider stimulus but rather as a supernormal one [98].

## Which characteristics of the respondents are predictors of fear and disgust rating of spiders and other arthropods?

Detailed analysis of the respondents' characteristics revealed that self-reported negative personal attitude toward spiders, high score in SPQ, and high score in DS-R reflected in more negative rating of all stimuli, particularly spider stimuli. Women also rated all stimuli, although spiders in particular, more negatively than men. Older respondents as well as those with biological type of education rated all stimuli more positively. This held true for ratings in both fear and disgust (see Fig 4). Although these results are generally in line with results of other researchers (negative emotions elicited by spiders [21, 32, 58, 60, 99–101]; gender differences [102]), two interesting points can be discussed.

First, SPQ scores themselves predicted mean rating given to spider stimuli by a given respondent (app. 60% of explained variability for both fear and disgust). This was expected to a certain degree–Mertens et al. [103], for example, found that specific sensitivity to fear of spiders, not general anxiety, was responsible for effective fear conditioning of participants in virtual reality experiments. Still, it is worth mentioning the high predictive value of the sensitivity to a specific fear of spiders alone. In fact, factors like gender or biological type of education, which are sometimes emphasized as very important (reviewed in [33]), proved to be very much secondary to this sensitivity represented by a simple SPQ score. This result can be of interest to clinical practitioners and other researcher when assembling, for example, terrain research or pilot studies.

Second, SPQ rather than DS-R provided a better predictor in disgust ratings. This is consistent with Sawchuk et al. [59] who found that spider phobics responded more with fear than disgust toward spider stimuli. However, other studies emphasize the importance of disgust in spider phobia as well [61, 101]. We contribute our result to DS-R questionnaire covering a broad spectrum of disgust-related questions whereas SPQ focusing specifically on spider and spider-like stimuli. Although DS-R can be divided into three theoretically independent subscales–core disgust, animal reminder disgust, and contamination-based disgust [67]–none of these subscales provided a significantly better prediction than the overall score. Perhaps this can be attributed to a specific position of spiders that can be perceived somewhere between the animal reminder disgust and contamination-based disgust. Alternatively, the testament of explicit SPQ simply overshadowed still quite broad orientation of DS-R subscales. This conclusion is supported by our first point as well.

## Is there a systematic difference in ratings of suspected phobic respondents compared to those with high, moderate, and low fear of spiders?

Owing to a relatively good sampling over the whole SPQ scale, we were able to define five categories that represented respondents with increasingly higher fear of spiders. We found that both fear and disgust mean scores of spider (Araneae) stimuli increased gradually with SPQ categories. This same, although less prominent trend was observed for other chelicerates and other arthropods. Although generally assumed, it was seldom shown on diverse groups of stimuli [21, 56] and/or respondents with diverse fear and disgust sensitivity [13, 14, 104].

When comparing different stimuli within the SPQ categories, other arthropods (insects, crustaceans, millipedes, and centipedes) were rated as eliciting the lowest fear by all SPQ categories. Accordingly, spiders and other chelicerates elicited higher fear in respondents of all SPQ categories. This result is crucial as it confirms our premise that spiders and spider-like chelicerates are more fear-eliciting than other groups of arthropods. To put it differently, spiders are a specific stimulus eliciting augmented fear in general population not just in people with high fear of spiders or in spider phobics. If this was not the case, the specificity of spiders could be doubted as a pathological deviation from standard (but see [105]). But according to our results, elevated fear of spiders compared to other arthropods is shared by all people in general, our work further points toward the evolutionary roots of negative emotions elicited by spider stimuli. To conclude, spiders are indeed special for everyone.

After validating the specificity of spider-like stimulus, we focused on differences between low fear and high fear respondents. Suspected phobic respondents scored insects, crustaceans, millipedes, and centipedes very similarly to respondents of almost all other categories (see Fig 5). In fact, if one category of respondents differed from the others, it would be respondents with extremely low fear of spiders. This was an important control confirming that high fear respondents were sensitive to specific fear of spiders, not general fear of all invertebrates or animals. Afterwards, we focused on spider and spider-like stimuli. In accordance with our expectations, suspected phobic respondents responded to them differently than respondents with low and extremely low fear of spiders. In behavioural tasks, similar results were previously reported for expectancy bias for encountering spiders [106], attentional bias to spider pictures [107], or stimulus-reaction task [108]. On the contrary, high fear respondents and suspected phobic respondents scored spider and spider-like stimuli very similarly (see Fig 5). In fact, exceptionally high correlations (95.3 and 92% of explained variability for fear and disgust, respectively) show that their scores were essentially the same. We confirmed that there was no difference in ratings of high fear and suspected phobic respondents (S15 Table in S1 File). Moreover, respondents with moderate fear (though their scoring was indeed somewhere in the middle) inclined more to the rating of the high fear and suspected phobic respondents than to that of low fear and extremely low fear respondents. Unexpectedly, it seems that the respondents with very low SPQ scores rather than suspected phobic ones deviate more from the average. To conclude, spiders are special but phobics not so much.

## General discussion

Our results show that spiders and spider-like chelicerates form a distinctive cognitive category but also that this category can be further split into two subcategories. The first one can be described as gracile spiders and spider-like chelicerates, the second as robust spiders. Since the robust spiders were generally the more frightening and disgusting stimuli, it could be argued that they form the core of the spider-like cognitive category. However, to the best of our knowledge, no spider species of this morphotype were relevant to human evolutionary history as a life-threatening stimulus. To our ancestors, only widow spiders of the genus *Latrodectus*

could have posed a real threat. In this sense, a smaller, not so robust morphotype would have been a better candidate to evolve into a prototypical spider stimulus.

Throughout the whole work, analyses based on fear ratings and disgust ratings provided very similar results. However, one important exception needs to be discussed. For all SPQ categories of respondents, spider-like stimuli elicit more fear than other arthropods. However, this is not true for disgust. Extremely low fear respondents rate spider-like stimuli (as a whole category) as less disgusting than other arthropods. To specify, the spider-like cognitive category is stable for all respondents but its relation to other arthropods on the disgust scale is different for a substantial section of our sample. However, fear is universal. It is further a typical feature of the whole spider-like category that they trigger more fear than disgust (see Fig 2 and previous section of Discussion.). Based on these results, we can argue that high fear is specific for spider-like category while high disgust is generally elicited by all arthropods. Evolutionary roots of the specificity of the spider-like stimulus should therefore be sought in fear, with disgust being only secondary of these two emotions.

Disgust is an emotion that prepares us to avoid infection in various behavioural tasks such as pathogen avoidance, mate choice, and social interactions [109, 110]. The categories of disgust elicitors are hence variable–parasites, vectors of diseases, body fluids, body injuries, hygiene threats, some sexual practices, and immoral acts [23]. The broad function of disgust led to the evolution of complex system of perceptual, emotional, and cognitive mechanisms that enable us to infer the potential infection risk. The resulting behavioural and physiological response protects the body from potential infection. This complex psychological and behavioural network is known also as behavioural immune system (BIS) [111]. Spiders are neither parasites, neither important vectors of human diseases [112–114]. Nevertheless, we can find other examples of generalization of pathogen disgust. Parasitic invertebrates are rated all highly disgusting [13, 33] but the same is true for insects [22] and some other non-parasitic arthropods in our study. The grater generalization of high disgust-eliciting stimuli should be adaptive for the complex task (to avoid all possible sources of infection) since false negative should be less costly than false positive in the case of BIS.

Nevertheless, it was already shown that the spider-like stimulus is simply not a spider of the order Araneae (see previous section of Discussion). We hypothesized that spiders might have represented a real threat to our ancestors thus a rapid fear response or at least a predisposition for fast associative learning of fear response [18, 103, 115, but see 40] would be highly adaptive, become genetically fixed and become non-associative fear [44, 116–118]. Here, we suggest extending this hypothesis on some other chelicerate stimuli, some of which are actually more dangerous to humans than extant spider species. Such chelicerates are, of course, scorpions [119]. Similar idea was already explored [120]. However, the scorpion was very clearly not a member of spider-like cognitive category in this study. Still, only one scorpion stimulus was included and therefore its true relation to spider-like cognitive category could not have been inspected in detail. For now, we cannot conclude on this question.

The second line of this study focused on investigating the effect of sensitivity to a specific fear of spiders on the perception of spider and spider-like stimuli. Rather unconventionally, we studied this effect across the whole SPQ scores scale. We found that the high fear respondents scored stimuli identically to suspected phobic respondents. The minimum SPQ limit to classify a respondent into a "high fear" category (SPQ > 15) was defined on the basis of an independent sample of Czech respondents (N = 3863) and it corresponds to 4th quartile of SPQ scores assessed from that survey [13, 14]. That means that about 25% of general population can be used very reliably as an approximation to truly phobic respondents who are much less prevalent in the population and often not comfortable with participation in this type of research. We cannot stress enough how important this result is to future arachnophobia

related research. It firstly significantly facilitates the recruitment of suitable respondents. In certain types of research, it secondly decreases a need for large samples of truly phobic respondents for whom such research may be emotionally demanding. We consider this the first of the two most important results of this study.

Multiple pieces of evidence can be named in support of evolutionary roots of negative emotions elicited by spider-like stimuli. They are the specificity of spiders among other invertebrates in general population (see previous section of Discussion), the high intensity of both fear and disgust they trigger [13, 31, 32], the existence of spider species which pose a real threat to humans [121], their association with pathogen disgust [20, 22], the results of visual attention tasks [42, 44], and the results of developmental studies [27–30]. Despite this fact, a simple and concise evolutionary explanation of negative emotions elicited by spider-like stimuli is difficult to formulate. We are aware that our study opens just as much questions as it answers. To further inspect possibility of evolutionary roots of spider-like cognitive category, we suggest addressing several issues in future research. First, all respondents in this work were Central Europeans, members of the so called WEIRD (Western, educated, industrial, rich, and democratic) society [122, 123]. Cross-cultural studies are needed to validate universality of discussed findings. Second, emotions elicited by live animals are rarely tested, yet live animals are the ultimate stimuli for evolution. In addition, animals' body size or motion are important characteristics of the stimuli [124] and therefore a study examining emotions elicited by live invertebrates is further needed. Third, although the spider-like cognitive category was relatively well explored in this work, other categories of fear- or disgust-eliciting invertebrates were not. A detailed comparison to other prominent groups of such invertebrates (e.g., scorpions) could shed more light into the research of animal phobias. Nonetheless, we consider the delineation of spider-like cognitive category the second of the two most important results of this study.

## Supporting information

**S1 Fig. Graphic results of the RDA analysis of stimuli's morphological traits.**
(TIF)

**S1 File.**
(XLSX)

## Acknowledgments

We would like to thank Mgr. Zuzana Štěrbová, Ph.D. for promoting of our research via social media and help with respondents recruiting. Furthermore, we thank all our participants for taking their time to complete the survey. Contribution of P.F. has been supported by Charles University Research Centre program No. 204069.

## Author Contributions

**Conceptualization:** Eva Landová, Daniel Frynta.

**Data curation:** Markéta Janovcová.

**Formal analysis:** Markéta Janovcová, Iveta Štolhoferová, Daniel Frynta.

**Funding acquisition:** Eva Landová.

**Investigation:** Eva Landová, Markéta Janovcová, Iveta Štolhoferová, Petra Frýdlová, Kristýna Sedláčková.

**Methodology:** Eva Landová, Silvie Rádlová, Daniel Frynta.

**Project administration:** Markéta Janovcová.

**Software:** Silvie Rádlová.

**Supervision:** Eva Landová, Daniel Frynta.

**Visualization:** Markéta Janovcová, Iveta Štolhoferová.

**Writing – original draft:** Eva Landová, Iveta Štolhoferová, Daniel Frynta.

**Writing – review & editing:** Eva Landová, Markéta Janovcová, Iveta Štolhoferová, Silvie Rádlová, Petra Frýdlová, Daniel Frynta.

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
