## [Decision Letter · Decision Letter 0]

29 Jun 2021

PONE-D-21-18124

Specificity of spiders among fear- and disgust-eliciting arthropods: Spiders are special, but phobics not so much

PLOS ONE

Dear Dr. Landová,

Thank you for submitting your manuscript to PLOS ONE. After careful consideration, we feel that it has merit but does not fully meet PLOS ONE’s publication criteria as it currently stands. Therefore, we invite you to submit a revised version of the manuscript that addresses the points raised during the review process.

Editorial comment: Two reviewers commented on your manuscript. As you can see from the reviews, both referees found the general topic addressed in your manuscript interesting and they provide a number of comments that might be helpful to further improve your work. Therefore, we invite you to submit a revision of the manuscript that addresses the remaining points together with a cover letter that contains point-by-point replies.

We look forward to receiving your revised manuscript.

Kind regards,

Michael B. Steinborn, PhD

Academic Editor

PLOS ONE

Journal Requirements:

Reviewers' comments:

Reviewer's Responses to Questions

**Comments to the Author**

1. Is the manuscript technically sound, and do the data support the conclusions?

Reviewer #1: Partly

Reviewer #2: Yes

2. Has the statistical analysis been performed appropriately and rigorously? 

Reviewer #1: I Don't Know

Reviewer #2: Yes

3. Have the authors made all data underlying the findings in their manuscript fully available?

Reviewer #1: No

Reviewer #2: Yes

4. Is the manuscript presented in an intelligible fashion and written in standard English?

Reviewer #1: Yes

Reviewer #2: Yes

5. Review Comments to the Author

Reviewer #1: Unfortunately, I was not able to assess the statistical analyses in detail as I was unable to review the supplements. In the text it states an Excel file summarizing details on statistics, however, even after consulting the Editor, no such file was available to me. Some of my remarks may thus ground on false believes about your data structure on my part. Please provide the missing files and data, so that the review process can be continued with all information necessary.

Kind regards

Reviewer #2: Background and Evaluation:

This study examines the sensation of fear and disgust for spider stimuli compared to other arthropods. In an online survey, participants evaluated a number of stimuli (spiders, non-spider chelicerates, and other arthropods) on a 7-point scale regarding the level of fear and disgust they experience. They found out that spiders as stimuli are evaluated higher on fear and disgust than other arthropods. In addition, a vast majority of spiders scored higher in fear than disgust, and vice versa for other chelicerates and arthropods. Results were similar among almost all different groups of responders, highlighting the evolutionary aspect of fear of spiders.

My overall evaluation is positive. Strength points include an interesting topic that is relevant to anxiety research. As a reader, the article provides a very logical explanation for the theoretical background (evolutionary significance), the findings and its implications. Additionally, the study has a big sample size. My suggestion is to improve the paper in regards of the theoretical background (SPQ vs FSQ and “cognitive category”) . My detailed comments are outlined below. Please note that my comments are aimed to further improving the manuscript, not to criticize the author’s work.

Line comments:

Page 3, line 48: The ‘fear of spider questionnaire’ is FSQ. The SPQ is called ‘spider phobia questionnaire’ (Klorman et al., 1974). Even though the questionnaires are similar in structure and content, they are different questionnaires (Szymanski & O’Donohue, 1995).

Page 4, line 6: “Related idea..” sentence is difficult to read.

Page 6, line 122: Please further explain “Do spiders form a single distinct cognitive category?”. I suggest either rewrite this question a little or provide more background as it is difficult to understand what “cognitive” means here. Some explanation is provided in the discussion section (page 20, lines: 440-448) and theoretical background (page 4, lines 84-85), however, I suggest to further elaborate “cognitive” category: One could interpret this as different emotional categories, and this would be very similar to research question #1.

6. PLOS authors have the option to publish the peer review history of their article (what does this mean?). If published, this will include your full peer review and any attached files.

Reviewer #1: No

Reviewer #2: No

---

## [Author Response · Author response to Decision Letter 0]

22 Aug 2021

Dear dr. Steinborn,

thank you for reviewing our manuscript. We have carefully studied the expert comments of both reviewers. We wrote a reply to each comment, or modified part of the manuscript according to the objections raised. We uploaded our answers as a "Responce_to_reviewers.docx" file. Furthermore, based on your requirements, we have adjusted the formal appearance of the manuscript to match the instructions of the journal. We've included the changes in the revised version of "Manuscript.docx" and also uploaded them in the tracked version ("Revised_Manuscript_with_Track_Changes.docx"). We have added a functional link to the data used in this work in the manuscript and reloaded the file with Supporting information (Reviewer 1 did not display the provided tables correctly). We believe that the modifications are in order and that the revised manuscript already meets the requirements of PLOS ONE magazine. 

Kind regards,

RNDr. Eva Landová, PhD

(corresponding author)

---

## [Decision Letter · Decision Letter 1]

9 Sep 2021

Specificity of spiders among fear- and disgust-eliciting arthropods: Spiders are special, but phobics not so much

PONE-D-21-18124R1

Dear Dr. Landová,

We’re pleased to inform you that your manuscript has been judged scientifically suitable for publication and will be formally accepted for publication once it meets all outstanding technical requirements.

Kind regards,

Michael B. Steinborn, PhD

Section Editor

PLOS ONE

Additional Editor Comments (optional):

Reviewers' comments:

Reviewer's Responses to Questions

**Comments to the Author**

1. If the authors have adequately addressed your comments raised in a previous round of review and you feel that this manuscript is now acceptable for publication, you may indicate that here to bypass the “Comments to the Author” section, enter your conflict of interest statement in the “Confidential to Editor” section, and submit your "Accept" recommendation.

Reviewer #1: All comments have been addressed

Reviewer #2: All comments have been addressed

2. Is the manuscript technically sound, and do the data support the conclusions?

Reviewer #1: Yes

Reviewer #2: Yes

3. Has the statistical analysis been performed appropriately and rigorously? 

Reviewer #1: Yes

Reviewer #2: Yes

4. Have the authors made all data underlying the findings in their manuscript fully available?

Reviewer #1: Yes

Reviewer #2: Yes

5. Is the manuscript presented in an intelligible fashion and written in standard English?

Reviewer #1: Yes

Reviewer #2: Yes

6. Review Comments to the Author

Reviewer #1: The authors provide novel hints towards the specificity of human responses to spiders in contrast to other (phenomenologically) similar species. The theoretical background and the methods sections were improved according to the reviewer’s suggestions. Furthermore, the authors also provided extensive information and explanations for a more detailed discussion of remarks made by the reviewers. Finally, since the authors provided the data used for analyses within the study, I was able to reproduce the part of the results of the paper (due to limited time, not every single calculation was tested).

I thus recommend accepting the manuscript in its current form even though minor changes in terms of proper English need to be made across the paper.

Reviewer #2: My overall evaluation is positive. In my opinion the authors sufficiently adressed the issues in the manuscript.

7. PLOS authors have the option to publish the peer review history of their article (what does this mean?). If published, this will include your full peer review and any attached files.

Reviewer #1: No

Reviewer #2: No

---

## [Editor Report · Acceptance letter]

14 Sep 2021

PONE-D-21-18124R1 

Specificity of spiders among fear- and disgust-eliciting arthropods: Spiders are special, but phobics not so much 

Dear Dr. Landová:

I'm pleased to inform you that your manuscript has been deemed suitable for publication in PLOS ONE. Congratulations! Your manuscript is now with our production department. 

Kind regards, 

on behalf of

Dr. Michael B. Steinborn 

Section Editor

PLOS ONE